# A Dynamic Protocol to Explore NLRP3 Inflammasome Activation in Cerebral Organoids

**DOI:** 10.3390/ijms25126335

**Published:** 2024-06-07

**Authors:** Dana El Soufi El Sabbagh, Liliana Attisano, Ana Cristina Andreazza, Alencar Kolinski Machado

**Affiliations:** 1Department of Pharmacology and Toxicology, University of Toronto, Toronto, ON M5S 1A8, Canada; dana.elsoufielsabbagh@mail.utoronto.ca (D.E.S.E.S.); alencarkolinski@gmail.com (A.K.M.); 2Krembil Brain Institute, Toronto Western Hospital, University Health Network, Toronto, ON M5T 2S8, Canada; 3Department of Biochemistry, University of Toronto, Toronto, ON M5S 1A8, Canada; liliana.attisano@utoronto.ca; 4Graduate Program in Nanosciences, Franciscan University, Santa Maria 97010-491, RS, Brazil

**Keywords:** NLRP3 inflammasome, cerebral organoids, neuroinflammation, 3D tissue, neuropsychiatric disease

## Abstract

The NLRP3 inflammasome plays a crucial role in the inflammatory response, reacting to pathogen-associated molecular patterns (PAMPs) and damage-associated molecular patterns (DAMPs). This response is essential for combating infections and restoring tissue homeostasis. However, chronic activation can lead to detrimental effects, particularly in neuropsychiatric and neurodegenerative diseases. Our study seeks to provide a method to effectively measure the NLRP3 inflammasome’s activation within cerebral organoids (COs), providing insights into the underlying pathophysiology of these conditions and enabling future studies to investigate the development of targeted therapies.

## 1. Introduction

### 1.1. The Nod-Like Receptor Pyrin Containing 3 (NLRP3)

The NLRP3 inflammasome is a complex protein structure comprising NLRP3, pro-caspase-1, and the adaptor molecule apoptosis-associated speck-like protein containing a CARD (ASC) [1,2]. This assembly is a crucial intracellular defense mechanism that is activated in response to potential threats, preventing cellular or tissue dysregulation [3]. Upon activation, the NLRP3 inflammasome triggers a cascade of immune responses: it initiates the release of activated caspase-1, which in turn prompts the secretion of pro-inflammatory cytokines interleukin (IL)-1β and IL-18 [4]. This series of reactions culminates in the production of additional inflammatory cytokines, including tumor necrosis factor (TNF)-alpha, interferon (INF)-gamma, and IL-6, thus amplifying the body’s inflammatory response to counteract the detected threat [5,6,7].

Extensive research has elucidated how the NLRP3 inflammasome is activated by pathogen-associated molecular patterns (PAMPs), including how bacteria, virus, and fungus can activate the NLRP3 inflammasome in immunocompetent subjects [8,9]. Similarly, damage-associated molecular patterns (DAMPs), including ATP, circulating cell-free mitochondrial DNA (ccf mtDNA), and hyaluronic acid release, which are indicators of cellular distress, are also known triggers for NLRP3 inflammasome activation [10,11,12]. Furthermore, recent studies have highlighted the role of oxidative stress in initiating an inflammatory response through the NLRP3 inflammasome. Specifically, high levels of reactive oxygen species (ROS), a type of DAMP, have been shown to bridge oxidative stress and inflammatory processes, underscoring the complex interplay between cellular damage signals and immune responses [13,14,15].

The inflammatory response serves as a vital defense mechanism for the organism, effectively combating infections and restoring tissue homeostasis after damage [16,17]. However, in certain conditions, the inflammatory response becomes chronic, resulting in detrimental effects outweighing its benefits [18]. Consequently, there is significant interest in comprehensively unraveling the underlying pathophysiology of conditions characterized by persistent inflammatory activation. Within this realm of investigation, the NLRP3 inflammasome emerges as a pivotal focal point for exploration [19].

### 1.2. Neuropsychiatric Diseases and the NLRP3 Inflammasome

Neuropsychiatric and neurodegenerative diseases, such as Alzheimer’s disease (AD), major depressive disorder (MDD), Parkinson’s disease (PD), and bipolar disorder (BD), are complex diseases posing a challenge for researchers attempting to comprehend their underlying mechanisms [20,21,22]. Growing evidence demonstrates that mitochondrial dysfunction and chronic inflammatory activation are pathways at the core of the pathophysiology of these illnesses [13]. described the relationship between mitochondrial complex I dysfunction and the inflammatory activation observed in patients with BD and indicate the NLRP3 inflammasome as the point of connection between these two impairments [19]. described the role of the NLRP3 inflammasome in pathophysiological changes of AD, showing that an increase in NLRP3 activation leads to a downstream cascade of proinflammatory cytokines which ultimately contributes to cognitive decline in AD patients [19]. The NLRP3 inflammasome is also involved in the pathogenesis of PD through two key proteins. The Parkin gene was found to regulate the inflammasome, and increased levels of alpha-synuclein aggregates were shown to increase NLRP3 activation [23]. Additionally, high levels of ccf mtDNA are strongly associated with the intense inflammation present in neurological diseases and can activate the NLRP3 inflammasome [24].

To further examine the inflammatory effects on individuals with neuropsychiatric disease, our team, Zhou et al. (2021) used peripheral blood mononuclear cells (PBMCs) from patients with BD and DD, and established a new protocol to activate the NLRP3 inflammasome in these cells through a lipopolysaccharide (LPS)-priming NLRP3 transcription stimulus followed by a nigericin NLRP3 inflammasome assembly signal [25]. After priming and assembly signals, ASC specks (a sign of NLRP3 inflammasome activation) were observed within cells via immunofluorescence microscopy. This technique allowed the authors to show that patients with BD are more sensitive to PAMPs/DAMPs exposure than control volunteers using peripheral samples, suggesting the possibility that this could also occur in the brain.

### 1.3. Use of Cerebral Organoids to Study the NLRP3 Inflammasome

Several studies have described inflammatory activation in post-mortem brain samples from patients with neuropsychiatric diseases [26,27,28,29]. However, these post-mortem samples might not fully capture the dynamic processes occurring in living tissue. Moreover, despite significant scientific progress, the lack of established protocols for assessing the NLRP3 inflammasome in a live brain remains a critical gap, impeding deeper insights into pathophysiology and hindering the development of targeted therapies. To bridge this gap, the generation of cerebral organoids (COs) derived from patients offers a promising avenue. These organoids can provide a more accurate and dynamic model of brain activity, potentially transforming our approach to understanding neuropsychiatric conditions and advancing drug discovery.

COs are three-dimensional (3D) self-organizing structures derived from embryonic stem cells (ESCs) or patient-derived induced pluripotent stem cells (iPSCs). COs comprise various neuronal cell types that come from an ectodermal lineage which develop into three-dimensional spheroid morphology [30,31,32,33]. COs enable researchers to test studies in a more complex tissue sample, in comparison to 2D neuronal cell culture. Being able to generate COs derived from patients’ cells or from established pluripotent stem cell lines enables the analysis of diverse context-dependent mechanisms of relevance to human diseases. Sivitilli et al. (2020) published an efficient protocol to generate human cerebral organoids (hCOs) from pluripotent stem cells with a rate of success 80% [32,33]. These organoids were composed of radial glial cells, including astrocytes, and mature neurons (Figure 1).

### 1.4. The NLRP3 Inflammasome: Astrocytes and Microglia

In the human brain, the NLRP3 inflammasome is present in both astrocytes and microglia, and both cell types have been used, together or independently, as models to study NLRP3 inflammasome activation [34,35,36]. While there are recently published protocols on generating microglia-containing COs (MCOs), they have not yet been standardized due to differences in neural induction and the complex patterning nature of 3D structures [37]. However, in COs astrocytes begin to develop at 12 weeks of age, with mature astrocytes arising after 24 weeks of culture, which is relevant for our study as astrocytes express the NLRP3 inflammasome [33]. Astrocytes are known to express NLRP3 and can activate the NLRP3 inflammasome [38] (Figure 1). Accordingly, understanding the innate role of NLRP3 in astrocytes is crucial for gaining insights into the mechanisms underlying neuropsychiatric diseases and exploring potential therapeutic interventions that target the inflammasome pathway. In a study by Freeman et al. (2017), for example, it was observed that the NLRP3 inflammasome could be activated in microglia and astrocytes upon exposure to lysophosphatidylcholine [39]. This activation resulted in ASC recruitment and activated caspase-1 release with both microglia and astrocytes acting as central players in terms of neuroinflammation activation. Additionally, Li et al. (2021) performed an in vivo study using an experimental model of depression based on NLRP3/Caspase-1/GSDMD-mediated pyroptosis and showed the role of the NLRP3 inflammasome in astrocytes. The study found astrocyte loss in the hippocampus, suggesting that NLRP3/Caspase-1/GSDMD-mediated pyroptosis is pivotal as a potential target for antidepressant drug screening [40]. In a prior study from our group, COs from ESCs and patient iPSCs were successfully generated, in which the patients’ genetic integrity was intact [41]. Inducing NLRP3 inflammasome activation in cell culture is a two-step process which we previously validated using a priming and activation step [25]. Specifically, the first step involves treatment with LPS that binds to a cell surface receptor to prime the cells and induce the transcriptional activation of NF-κB. Following priming, nigericin or ATP are used in the second step to assemble the NLRP3 inflammasome complex composed of the NLRP3 protein, pro-caspase 1, and ASC (Figure 2). The use of NLRP3 inflammasome inhibitors is also important to perform artificial studies on the NLRP3 modulation. There are several known NLRP3 inflammasome blockers, such as MCC950, CY-09, tranilast, oridonin, 3,4-methylenedioxyb-nitrostyrene, and OLT1177 [42,43,44,45,46,47,48]. MCC950 is one of the most described NLRP3 inhibitors in the literature. Due to its capacity of binding the active and inactive forms of the NLRP3, MCC950 is considered a classic inhibitor of this mechanism since it can avoid the NLRP3 inflammasome oligomerization. As a negative control, MCC950 is applied after priming but before activation [40].

Conventional protocols designed for 2D cell culture have limited applicability to 3D tissue culture, often resulting in disparate outcomes. With the growing adoption of COs and other 3D spheroids, there is a pressing need to validate protocols specifically tailored for 3D tissue environments. Thus, the aim of this study was to develop a protocol, modified from Zhou et al. (2021), to model NLRP3 inflammasome activation in 3D COs generated from ESCs and to use this method in iPSC-derived COs [25]. Here, a time-response curve was generated using three different study groups to identify the optimal incubation time for nigericin, the NLRP3 inflammasome activator, while the primer LPS was kept constant.

## 2. Materials and Methods

### 2.1. Methodological Experimental Design

This protocol aims to activate the NLRP3 inflammasome in COs and to capture its activation by detecting ASC specks. Initially, ESCs were cultured and used to generate COs which were sliced at 6 months of age. The LPS-priming signal followed by nigericin exposure was performed at different time points. Immunofluorescence analysis was then carried out to detect ASC specks. The production of nitric oxide (NO), total levels of ROS, and the release of dsDNA, IL-1β, ccf mtDNA, and caspase-1 were assessed via colorimetric and fluorometric assays. All experiments were performed in triplicate.

### 2.2. Protocol to Activate the NLRP3 Inflammasome in COs

#### 2.2.1. Generation and Long-Term Culture of COs


Cerebral organoids (COs) were generated from H9 ESCs using an established protocol at the Applied Organoid Core Facility (ApOC) at the Donnelly Centre, University of Toronto) [32].COs were cultured on an orbital shaker in six-well non-tissue culture plates using 3 mL of maturation medium which was changed twice a week.Once CO size exceeded 2 mm in diameter, a maximum of four COs were placed in each well and allowed to grow to 6 months of age (24 weeks).


#### 2.2.2. Slicing COs with the Vibratome


COs were placed in plastic base molds, and excess media was removed to allow the CO to dry.Low-melting-point 4% agarose was warmed in PBS to 37 °C, and the CO was embedded in agarose and carefully centered in the base mold.Base molds were placed on ice for 10 min.Once solidified, the CO embedded in agarose was scooped out and cut into a 1 cm ±× 1 cm cube.A drop of glue was placed on the vibratome’s stage block, and the CO cube was carefully placed on it.The vibratome’s stage was submerged in the block filled with cold PBS+/+ (with calcium and magnesium). This helped keep the slice alive and reduce tissue damage.The vibratome was then lowered and moved to allow the blade to meet the CO for slicing into thin sections.Each CO was sliced at a speed of 1 mm/sec and an amplitude of 0.8 to generate 400 µm thick sections. This size allows for better drug penetration to all cell types in the CO while maintaining CO structure and integrity.Sliced sections were collected from the vibratome block using a transfer pipette and were placed in a six-well plate containing PBS and 1% penicillin/streptomycin.After slicing, the plate was placed in the biosafety cabinet (BSC), and the CO slices were transferred to a new six-well plate pre-filled with 2 mL CO maturation media with 1% penicillin/streptomycin. This avoids contamination as the CO slices had been exposed to nonsterile conditions during slicing.


#### 2.2.3. Activating the NLRP3 Inflammasome


Note: Three groups were used to generate a time-response curve of the NLRP3 activation protocol as depicted in Figure 3.The original protocol (used for PBMCs) involved priming using 100 ng/mL LPS (Invivogen, San Diego, CA, USA, TLRL-3PELPS LPS) for 3 h, followed by pre-treatment with an NLRP3 inhibitor at 100 nM MCC950 (Invivogen, San Diego, CA, USA, 210826-40-7) for 2 h and then 10 µM of nigericin (Sigma, St. Louis, MO, USA, N7143), an NLRP3 activator for 1 h.Here, the LPS and MCC950 incubation times were consistent with the original protocol; however, the nigericin incubation time was altered. All treatments were performed using a shaker during the incubations.Slices were primed with 100 ng/mL LPS for 3 h at 37 °C.Media was removed, and slices were then incubated at 37 °C with 100 nM MCC950 for 2 h as a pre-treatment.Media was removed, and slices were incubated with 10 µM nigericin for 1, 4, and 16 h for the time-response curve.Following nigericin incubation, the supernatants were collected and used to assess the levels of NO, ROS, dsDNA release, and IL-1β and caspase 1 expression as described in Section 2.3.


#### 2.2.4. Fixing of CO Slices and Immunostaining


After the removal of supernatants, slices were fixed with 4% paraformaldehyde (PFA) overnight, then transferred to 30% sucrose for another overnight incubation, and finally embedded in OCT gel at −80 °C and then cryosectioned at 20 µm thick for immunofluorescence staining.Cryosections were stained as previously described by Duong et al. (2021) using GFAP for astrocytes (Abcam, Cambridge, UK, # ab4674), MAP2 for neurons (I3-1500), SOX2 for neural progenitors (MAB2018), and ASC for ASC specks (AL177) antibodies to examine ASC speck formation in astrocytes within each CO slice [41]. Secondary antibodies used were Goat anti-Mouse [IgG] [H + L] (Abcam, #Ab97035) secondary antibody Cy3, Donkey anti-Rabbit IgG [H + L] secondary antibody Alexa Fluor Plus 647, (Invitrogen, #A32795), and Donkey anti-Chicken IgY [H + L] secondary antibody Alexa Fluor 488 (Jackson ImmunoResearch Laboratories Inc., West Grove, PA, USA, #703-546-155). All antibodies were diluted using 0.5% BSA, and following staining, coverslips were mounted using ProLong Gold Antifade Mountant with DAPI (Invitrogen, Waltham, MA, USA, P36935), and samples were imaged at the Microscopy Imaging Laboratory, University of Toronto, Canada, using the LSM 880 Elyra Superresolution confocal microscope.


### 2.3. Developed Assays of Parameters Related to Inflammatory Activation

#### 2.3.1. Indirect Measurement of Nitric Oxide

Nitric oxide (NO) production was assessed in supernatant samples using an indirect technique published by Choi et al. (2012) [49]. This method is based on the use of *Greiss* reagent, which allows for the detection of nitrite (NO_2_^−^) and metabolic nitrate. Absorbance was measured using a plate reader using Gen5 Software (BioTek Instruments, Inc., Winooski, VT, USA, 253147) at 540 nm. As this is a semi-quantitative assay, the results were expressed as percentage compared to the negative control, where untreated organoids were considered 100% in the calculation.

#### 2.3.2. Semi-Quantitative Analysis of Total Levels of Reactive Oxygen Species

Total levels of ROS were determined using a fluorometric assay based on the 2′,7′-dichlorofluorescin diacetate (DCFH-DA) reagent as described by Costa et al. (2012) [50]. DCFH-DA is deacetylated to dichlorofluorescin (DCFH), which when in contact with ROS is converted to dichlorodihydrofluorescein (DCF), a fluorescent molecule. Fluorescence intensity was determined at 488 nm of excitation and 525 nm of emission in a plate reader using Gen5 Software (BioTek Instruments, Inc., Winooski, VT, USA, 253147). As this is a semi-quantitative assay, the results were expressed as percentage compared to the negative control, where untreated organoids were considered 100% in the calculation.

#### 2.3.3. Measurement of Extracellular dsDNA

The quantification of extracellular dsDNA in the supernatant was performed using a Quant-iTTM PicoGreen^®^ kit (ThermoFischer, Waltham, MA, USA, P11495) as described by Ahn, Costa, and Emanuel (1996) [51]. PicoGreen intercalates between dsDNA molecules and then emits fluorescence, allowing for the assessment of cell integrity. Fluorescence was determined at 480 nm excitation and 520 nm emission in a plate reader using Gen5 Softwareer (BioTek Instruments, Inc., Winooski, VT, USA, 253147). As this is a semi-quantitative assay, the results were expressed as percentage compared to the negative control, where untreated organoids were considered 100% in the calculation.

#### 2.3.4. Interleukin-1β and Caspase-1 Protein Production

Activated human IL-1β production was measured using an Elisa kit (Abcam, cat. 2ab214025, Cambridge, UK) following the manufacturer’s instructions. Results were expressed as pg/mL.

Activated caspase-1 production was assessed by using an in-house previously published protocol [25]. Supernatant (50 µL) was added to a high-binding 96-well plate (Greiner, Monroe, NC, USA, #655061) and incubated on a shaker at 4 °C overnight. After blocking for 1 h with a bovine serum albumin blocking solution (5% BSA), the caspase-1 primary antibody was added for 1 h (Abcam, ab62698) followed by a secondary antibody (anti-rabbit IgG, HRP-linked antibody, 7074S, Cell Signaling Technology, Danvers, MA, USA) for 1 h. Next, 3,3′,5,5′-Tetramethylbenzidine (TMB) was used as the chromogenic substrate for 15 min, and 100 μL of 1 M HCl was used to stop the reaction. Absorbance was determined at 450 and 540 nm in a plate reader using Gen5 Software (BioTek Instruments, Inc. 253147). Results were calculated as percentage compared to the negative control.

#### 2.3.5. Circulating Cell-Free Mitochondrial DNA Measurement

Levels of ccf mtDNA were measured as described in Lu et al. (2024) [52]. Cell supernatants (100 µL) were used to extract mitochondrial DNA using the QiaAMP DNA mini kit. Extracted mtDNA was mixed with 50 µL ultra-pure DNAse-free and RNAase-free water [52]. Commercial oligonucleotide was serially diluted to a concentration ranging from 108 to 102 copies/µL to estimate the absolute concentration of ccf-mtDNA. The ND4 and ND1 mitochondrial genes and the B2M and PPIA nucleon gene were used for the gene expression. TaqManTM Duplex PCR was performed on BioRad’s C1000 Thermal cycle CFX96 Real Time System using 20 µL of reaction mixture including 10 µL TaqManTM Fast Advanced Master Mix (ThermoFischer, 4444556), 4 µL of mtDNA sample, 1 µL of each primer (forward and reverse), and 1 µL TaqManTM probe (each gene). qPCR cycling conditions were: 50 °C for 2 min, 95 °C for 20 s, 40 cycles of 95 °C for 3 s, and 60 °C for 30 s. Results were expressed as ccf-mtDNA copies/µL. See Table 1 for list of PCR products used.

### 2.4. Image Analysis

In our study, we began by quantifying glial fibrillary acidic protein (GFA) cells, DAPI stained nuclei, and ASC positive cells in the CO sections. This quantification was performed using Indica Lab’s HALO FL CytoNuclear image analysis software (Version 3.5.3577.265), which allows for accurate and automated cell counting and characterization. Representative images were used to train the algorithm in HALO to recognize and differentiate between GFAP, DAPI, and ASC positive cells based on their staining pattern and morphology. This included setting parameters for nuclear size, intensity thresholds, and cytoplasmic intensity, all optimized to accurately identify the segment cells of interest while minimizing background noise and false positives. This algorithm generated masks for DAPI, GFAP, and ASC positive cells, and quantitative data were extracted yielding GFAP+, ASC+, and total cells (DAPI).

While HALO CytoNuclear (Version 3.5.3577.265) successfully quantified ASC+ cells, it was unable to count the number of ASC specks on each cell due to their significantly small size. Thus, we designed a semi-automated program on FIJI (ImageJ) (version 2.9.0/1.53t) to detect ASC specks based on their size, circularity, and brightness. This included several parameters: size threshold to exclude non-speck objects, circularity to ensure the roundness of specks, and brightness to differentiate specks from the background. Since this was a semi-automated program, after applying the macro to the images, each detected speck was manually reviewed to ensure full accuracy. This combined approach allowed for a precise detection of ASC specks which were too small to be reliably counted using HALO CytoNuclear. Images were re-labeled with random numbers and were validated by a second observer to remove potential bias.

### 2.5. Statistical Analysis

Results of the measured parameters related to inflammatory activation were first plotted in Microsoft Excel (Version 16.77.1) and then converted to percentage relative to the negative control. Data were analyzed by One-way Anova followed by Tukey post hoc. Comparisons with *p* < 0.05 were considered significant.

## 3. Results

Our results described below demonstrate the effective activation of the NLRP3 inflammasome in cerebral organoids. This study validates our modified protocol, that reflects the necessity of tailoring experimental approaches to 3D cultures and highlights the potential use of cerebral organoids for advancing our understanding of brain diseases.

### 3.1. Results of Time-Response Curve (Groups I, II, and III)

The treatment of CO slices with LPS and varying times of Nigericin incubation enabled us to determine the optimal treatment time for 3D tissues. Figure 4 shows each group of CO slices stained for neural progenitor cells (SOX2), astrocytes (GFAP), and ASC specks. Each group included an untreated control slice, an activation slice (LPS+ nigericin (X h)), and an inhibition slice (LPS + MCC950 + nigericin (X h), bottom image). The images on the top row are the activation slice and are used to compare to the respective inhibitory slice (bottom row) of each group. In group 1 (Figure 4A) very few specks emerge following activation with LPS + 1 h of nigericin. In group II (Figure 4B) CO slices were treated with 4 h of nigericin, and the emergence of ASC specks (top left images) is much more pronounced in comparison to the LPS + MCC950 + Nigericin negative control (bottom left images). Group 3, with the 16 h incubation of nigericin, showed very few specks emerging like group I (Figure 4C). The quantitation of images using HALO and the ImageJ software confirmed that group II (LPS + Nigericin) treatment showed a significant increase in the number of specks as compared to 1 h or 16 h treated samples (Figure 4D). To further verify the ASC specks, images were quantified for ASC positive (ASC+) and GFAP positive (GFAP+) cells using HALO, and only cells that were double positive were used in ASC speck quantification.

Following the initial results, the group II procedure was repeated and slices immunostained using MAP2, a neuronal cell surface marker, as well as GFAP and as ASC specks (Figure 5). These markers were specifically selected to further validate that the ASC specks are only located on the astrocytes, not on other neuronal cell types within the COs. In Figure 5A, the CO slice was treated with LPS + 4 h nigericin, and ASC specks are seen on the cells, while few specks are seen in the LPS + MCC + Nigericin slice and control untreated slice. Figure 5B shows ASC speck quantification in the treatment groups where the LPS + Nigericin group shows a higher number of ASC specks in comparison to the untreated and LPS + MCC950 + Nigericin groups.

### 3.2. Validating NLRP3 Inflammasome Activation

Supernatant from each CO slice was collected and used to perform assays to examine the effect of NLRP3 inflammasome activation on cellular health, by considering parameters related to inflammatory activation. Figure 6 shows the results obtained from samples of Group II, where nigericin was used as the activator for 4 h. Figure 6A–C show a more specific measurement of NLRP3 inflammasome activation as ccf mtDNA, caspase-1, and IL-1β levels, respectively, were measured. As expected, levels of ccf mtDNA, caspase-1, and IL-1β were significantly increased in the activation slice, while they were significantly decreased in the inhibition slice. Levels of NO production were also measured (Figure 6D) and show that the activation slice (LPS + Nigericin) had a significant increase in NO as compared to the control. On the other hand, a significant decrease in NO was observed in the inhibition group (LPS + Nigericin + MCC950). Similarly, in Figure 6E,F levels of ROS and dsDNA release, respectively, were measured. ROS levels and extracellular dsDNA amounts were significantly increased in the activation slice, while they were significantly decreased in the inhibition slice.

## 4. Discussion

Our team has made significant strides in understanding the activation mechanisms of the NLRP3 inflammasome within cell models to probe the roots of neuroinflammation. Notably, our methods, particularly the application of immunofluorescence to detect intracellular ASC speck formation, have been crucial in illuminating this process, as outlined in our recent work [25]. Building on this, we have refined our protocols for better utility in the context of the unique characteristics of 3D cerebral organoids (COs). We focussed on determining the best incubation times for treatments nigericin, as the activator of the NLPR3 inflammasome, to accommodate the thicker structure of COs compared to traditional 2D cell cultures.

It is important to mention that in addition to NLRP3, there are other inflammasomes well described in the literature, such as NLRP1, NLRC4, and AIM2, for example. Fenini (2018) have used UVB and nigericin to activate NLRP1 in keratinocytes [53]. Rozario et al. (2024) also described the use of depleting cytosolic potassium ions inducers, such as nigericin, to activate the NLRP1 inflammasome in nonhematopoietic cells [54]. However, there is still a lack of robust data on whether this process would also occur in CNS cells. Additionally, to the extent of our knowledge, there is no scientific information about nigericin inducing NLRC4 or AIM2 inflammasome activation, which actually can be promoted by NAIPs sensing type 3 secretion system and poly(deoxyadenylic-deoxythymidylic acid sodium salt, respectively [38,55]. In this regard, we strongly believe that this protocol will specifically induce NLRP3 inflammasome activation, through priming and assembly signals.

While NLRP3 activation mechanisms are well-established to be triggered by both pathogen-related and damage-related signals, its role in persistent inflammatory states, especially in neuropsychiatric disorders like bipolar disorder (BD), is still being unraveled [13]. The advent of 3D COs derived from patient cells offers a more physiologically relevant model than post-mortem samples or standard 2D cell culture and have the potential to enhance our understanding of disease pathophysiology and inform therapeutic development. However, this model has limitations, given the absence of microglia in the COs due to current technical constraints, which narrows our perspective to astrocyte-based inflammation. Despite this, the observed activation patterns within these COs can provide valuable insights into the neuroinflammatory process, validated by changes in markers like nitric oxide (NO) levels, reactive oxygen species (ROS), and the release of inflammatory cytokines and ccf mtDNA.

This work underscores the necessity of adapting research methodologies to suit the complexities of 3D biological models. It also highlights that protocols effective in 2D cultures may require significant adjustments for application in more complex 3D systems. Our findings, while preliminary, represent a step forward in the quest to decipher the intricate dance of neuroinflammation, particularly as it pertains to the role of the NLRP3 inflammasome in mental health disorders. It is also important to mention that this study is the opening door to performing further pharmacological investigations to develop new anti-neuroinflammatory agents through NLRP3 modulation. There are many well-known NLRP3 inflammasome inhibitors, such as MCC950, CY-09, tranilast, oridonin, 3,4-methylenedioxyb-nitrostyrene, and OLT1177 [42,44,45,46,47,48]. However, when considered as potential anti-inflammatory therapies, these molecules may cause challenging side effects [56]. In this regard, many research groups have been looking forward to new NLRP3 modulators, and natural health products are highlighted in this field of investigation. Our group has also described a potential NLRP3 inflammasome blocker which is an extract derived from an Amazon rainforest berry popularly known as açaí [15,57]. Perhaps this natural health product could be tested through the new protocol described here.

Moving forward, it is essential to refine the generation of COs to include microglial components, to enhance fidelity to brain tissue. Although challenges remain, our progress lays the groundwork for future studies aiming at dissect the nuanced mechanisms of neuroinflammation, paving the way for targeted, patient-specific treatments.

## Figures and Tables

**Figure 1 ijms-25-06335-f001:**
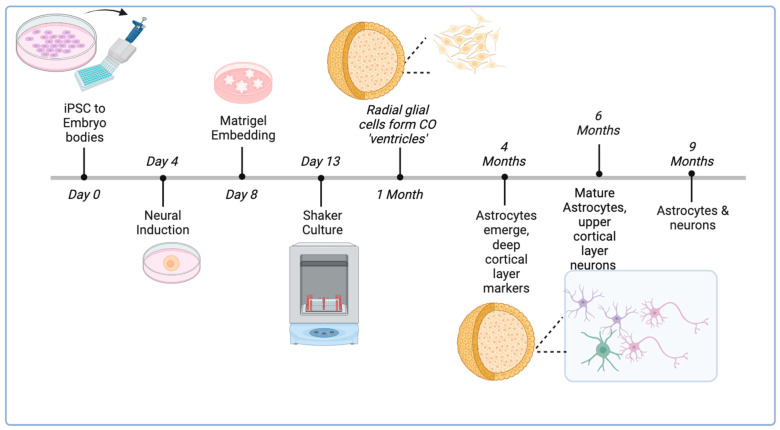
Overview of CO generation protocol and long-term growth. ESCs or iPSCs are singularized and plated to form embryo bodies and then grown into a neural induction phase. Once the 2D spheres are intact, organoids are embedded in Matrigel to form 3D structures and then excised and placed in rotating cultures that grow into uniform spheroids. At the 1-month time point, they primarily contain radial glia cells, while at 4 months and 6 months, astrocyte progenitors emerge. After 6 months, COs contain mature deep-layer and upper-layer neurons, mature astrocytes.

**Figure 2 ijms-25-06335-f002:**
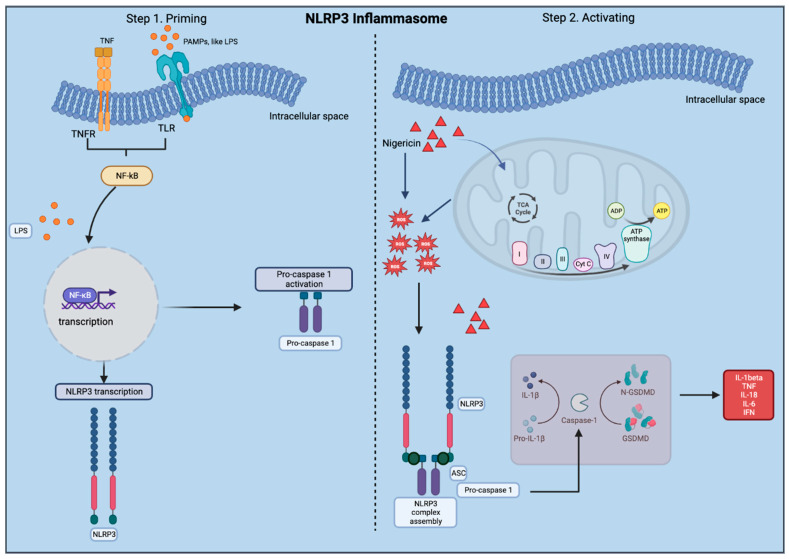
Overview of the NLRP3 inflammasome. Step 1 is a priming process, and step 2 involves the activation of the inflammasome which results in increased inflammatory cytokine production.

**Figure 3 ijms-25-06335-f003:**
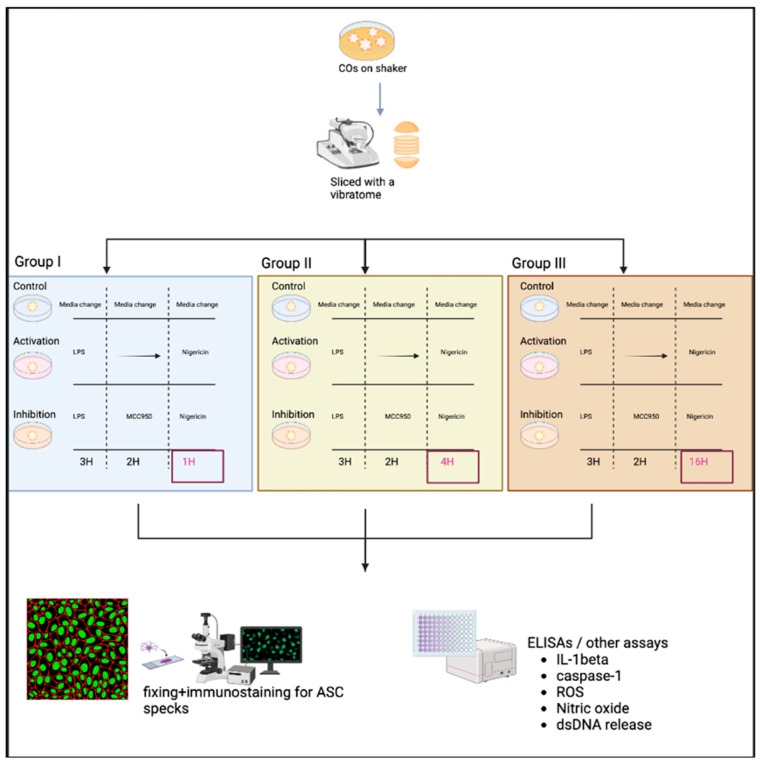
The experimental scheme of the time-response curve in CO slices in groups I, II, and III. After the LPS-priming stimulus and MCC950 exposures, group I was treated with nigericin for 1 h, group II for 4 h, and group III for 16 h. Slices were then fixed and immunostained, while the supernatant was collected for various assays of inflammatory activation.

**Figure 4 ijms-25-06335-f004:**
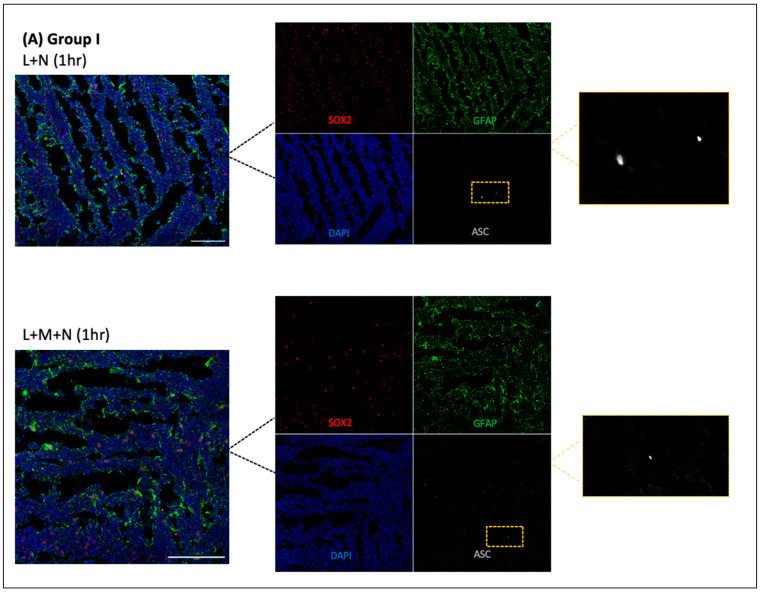
Time-response curve of the LPS + Nigericin protocol using groups I, II, and III. (**A**) Group I: 3 h LPS treatment followed by 1 h Nigericin treatment (**top**). 3 h LPS treatment, 2 h MCC950 pre-treatment, and 1 h Nigericin treatment (**bottom**). (**B**) Group II: 3 h LPS treatment followed by 4 h Nigericin treatment (**top**). 3 h LPS treatment, 2 h MCC950 pre-treatment, and 4 h Nigericin treatment (**bottom**). (**C**) Group III: 3 h LPS treatment followed by 16 h Nigericin treatment (**top**). 3 h LPS treatment, 2 h MCC950 pre-treatment, and 16 h Nigericin treatment (**bottom**). Slices were stained with SOX2 (neural progenitor cells), GFAP (astrocytes), and ASC (ASC speck formation in the NLRP3). Scale bar = 100 µm. (**D**) By counting cells with HALO and ASC specks with ImageJ. Results are shown as a ratio of cells expressing ASC specks in each group. Data were analyzed by two-way ANOVA with Sidak’s multiple comparison test. * *p* < 0.05. (**E**) most cells containing ASC specks in group II were found to be double positive ASC+/GFAP+.

**Figure 5 ijms-25-06335-f005:**
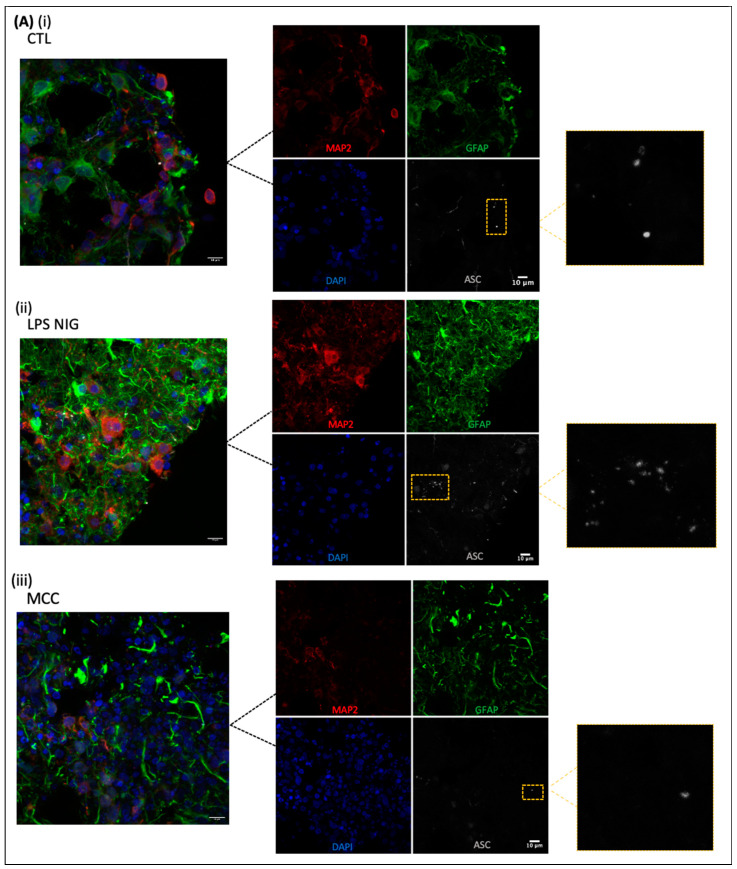
Method reproducibility for group II: Locating the ASC specks in astrocytes. (**A**) (**i**) Control slice. (**ii**) LPS (3 h) + Nigericin (4 h). (**iii**) LPS (3 h) + MCC950 (2 h) + Nigericin (4 h). Staining performed using MAP2 (neurons) GFAP (astrocytes) ASC (ASC activation in NLRP3). Scale bar = 10 µm. (**B**) Images quantified for ASC specks using HALO and ImageJ. Data were analyzed by two-way ANOVA with Sidak’s multiple comparison test. * *p* < 0.05 CTL vs. L + N and # *p* < 0.05 L + N vs. L + M + N.

**Figure 6 ijms-25-06335-f006:**
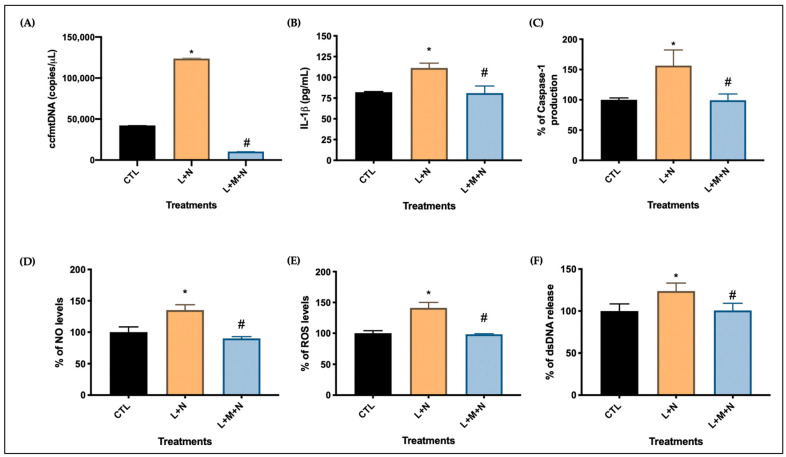
Validation on NLRP3 inflammasome endpoints of group II. (**A**) Measurement of ccf mtDNA; (**B**) IL-1β protein expression analysis; (**C**) Caspase-1 protein expression; (**D**) Indirect measurement of NO levels; (**E**) Semi-quantitative evaluation of ROS levels. (**F**) Measurement of dsDNA release. Data in (**A**–**F**) are expressed as means and standard deviation of three technical replicates of a single experiment. Statistical analyses were performed using a One-way ANOVA to examine differences followed by Tukey’s multiple comparison test (* *p* < 0.005 CTL vs. L + N and # *p* < 0.005 L + N vs. L + M + N).

**Table 1 ijms-25-06335-t001:** Primers used for measuring ccf mtDNA.

PCR Primers for ccf mtDNA	
ND4 F	F1 5′-CCATTCTCCTCCTATCCCTCAAC-3′
ND4 R	5′-ACAATCTGATGTTTTGGTTAAACTATATTT-3′
ND4 Probe	5′-FAM/CCGACATCA/ZEN/TTACCGGGTTTTCCTCTTG/3IABkFQ/-3′
ND1 F	5′-CCCTAAAACCCGCCACATCT-3′
ND1 R	5′-GAGCGATGGTGAGAGCTAAGGT-3′
ND1 Probe	5′-HEX/CCATCACCC/ZEN/TCTACATCACCGCCC/3IABkFQ/-3′
ND4 + ND1 geneblock	CACGAGAAAACACCCTCATGTTCATACACCTATCCCCCATTCTCCTCCTATC CCTCAACCCCGACATCATTACCGGGTTTTCCTCTTGTAAATATAGTTTAACCA AAACATCAGATTGTGAATCTGACAACAGAGGCTCTCTTCACCAAAGAGCCC CTAAAACCCGCCACATCTACCATCACCCTCTACATCACCGCCCCGACCTTAG CTCTCACCATCGCTCTTCTACT ATGAACCCCCCTCCCCATACCCAA-3′
B2M F594	5′-TGCTGTCTCCATGTTTGATGTATCT-3′
B2M R679	5′-TCTCTGCTCCCCACCTCTAAGT-3′
B2M-Probe	5′-FAM/TTGCTCCAC/ZEN/AGGTAGCTCTAGGAGG/3IABkFQ/-3′
PPIA-F	5′-GTGGCGGATTTGATCATTTGG-3′
PPIA-R	5′-CAAGACTGAGATGCACAAGTG-3′
PPIA Probe	5′-/56-FAM/AAT TCA CGC/ZEN/AGA AGGA ACC AGA CAG T/3IABkFQ/-3′

## Data Availability

Data is available under request.

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
