# Peer review of "A Dynamic Protocol to Explore NLRP3 Inflammasome Activation in Cerebral Organoids"

_ijms, 2024, doi:10.3390/ijms25126335_

Round 1

Reviewer 1 Report

Comments and Suggestions for Authors

The review by Sabbagh et al., seeks to develop a protocol for NLRP3 inflammasome induction using cerebral organoids (CO).  The protocol developed in this manuscript is a modification of a standard method that is used in the inflammasome field involving LPS

priming followed by nigericin treatment.  Therefore, the assay is not novel but a modification of standard methods in the field.  In addition, the CO appear to contain predominantly astrocytes and labeling methods to characterize other CNS cell types are lacking.  Lastly, the quantification of cytokines, caspases and other NLRP3 endpoints (Figure 6) need to be modified.

Major Points:

1.        The protocol developed is not novel, but a modification of standard methods in the field. Therefore, the contribution to the field in minimal.

2.        The characterization of CNS cell types in the CO needs to be expanded.  Other cell type markers that define microglia, neurons and oligodendrocytes should be included in the study.

3.         All suppliers and catalog numbers for reagents used in this study should be added to the Methods.

4.        Methods for staining ASC specks and quantification are lacking in the manuscript.

5.        Data in Figure 6 shows endpoints of inflammasome products.  The data is presented as percentages.  The exact values of each test should be provided.  For example, what are the levels of IL-1 beta in Figure 6B?

6.        The author should include a Discussion of other inflammasome that maybe activated by this protocol. 

Comments on the Quality of English Language

acceptable

Reviewer 2 Report

Comments and Suggestions for Authors

The present study evaluates methods for measurement of NLRP3 inflammasome activation within cerebral organoids. This will provide insights into the underlying biologic mechanisms of these conditions and provide knowledge used in the development of future targeted therapies.

General aspects

The activation of the NLRP3 inflammasome plays a crucial role in several degenerative diseases, including the human brain. The manuscript is well written and contains a detailed protocol for generation and activation of the NLRP3 inflammasome in a sophisticated organoid model. From my point of view the importance of the manuscript should be substantial improved if also result from potential inhibitors for the NLRP3 inflammasome activation ere included. However, if the journal support publication of pure study protocol I suggest that the paper could be published after only a few corrections and adjustments.

Specific points:

Line 1               Could Article be replaced with Protocol?

Line 4               Should it be a 5th author after and?

Line 28             Add a specific reference after IL-18.

Line 128          Add information and references about potential inhibitors for NLRP3 activation.

Line 159-219 Format text in accordance to the rest of the manuscript.

Line 247          Change expression to production. Expression is associated with mRNA levels. Include the world activation for both IL-1β and caspase-1. Both proteins ar produced as precursors that is stored inactive intracellularly, before activation and cleavage through NLRP3 activation.

Line 271-290 Put all the primer sequences int o table with an explanatory table head.               

Line 304-378 Include subheading in the result section in line with that in the materials and methods section.

Line 407          Discuss potential NLRP3 inhibitors that can be applied in the present model system.

Line 419-426 Suggest the authors should add also some more recent references.

Round 2

Reviewer 1 Report

Comments and Suggestions for Authors

Authors have adequately addressed concerns.